# Susceptibility Genes Associated with Multiple Primary Cancers

**DOI:** 10.3390/cancers15245788

**Published:** 2023-12-10

**Authors:** Mengyao Lu, Xuemei Zhang, Qian Chu, Yuan Chen, Peng Zhang

**Affiliations:** Department of Oncology, Tongji Hospital, Tongji Medical College, Huazhong University of Science and Technology, Wuhan 430030, China; m202176432@hust.edu.cn (M.L.);

**Keywords:** multiple primary cancers, susceptibility genes, genetic predisposition, second cancer, next-generation sequencing

## Abstract

**Simple Summary:**

With advanced cancer treatments and screening techniques, a growing number of cancer survivors face the challenge of multiple primary cancers (MPCs), affecting approximately one in six patients. Distinguishing MPCs from recurrences or metastases is still challenging due to the limited clinicopathological criteria available. Next-generation sequencing (NGS) has facilitated the comprehension of MPCs from a genetic standpoint. In this review, we discuss three main inherit factors contributing to MPCs development: the mutations targeting oncogenes and caretaker genes directly involved in MPCs, the potential influence of pleiotropic loci spanning various cancer types, and the regulatory impact of deleterious mutations following treatment. These susceptibility genes will be discussed in relation to their background in signaling pathways. Additionally, the association network between genetic signatures and various tumor pairs will be summarized.

**Abstract:**

With advancements in treatment and screening techniques, we have been witnessing an era where more cancer survivors harbor multiple primary cancers (MPCs), affecting approximately one in six patients. Identifying MPCs is crucial for tumor staging and subsequent treatment choices. However, the current clinicopathological criteria for clinical application are limited and insufficient, making it challenging to differentiate them from recurrences or metastases. The emergence of next-generation sequencing (NGS) technology has provided a genetic perspective for defining multiple primary cancers. Researchers have found that, when considering multiple tumor pairs, it is crucial not only to examine well-known essential mutations like MLH1/MSH2, EGFR, PTEN, BRCA1/2, CHEK2, and TP53 mutations but also to explore certain pleiotropic loci. Moreover, specific deleterious mutations may serve as regulatory factors in second cancer development following treatment. This review aims to discuss these susceptibility genes and provide an explanation of their functions based on the signaling pathway background. Additionally, the association network between genetic signatures and different tumor pairs will be summarized.

## 1. Introduction

Cancer is a leading cause of death worldwide [1]. However, advancements in screening techniques and treatments have significantly improved cancer patient outcomes, with approximately 70% of individuals surviving for over five years [2]. The number of cancer survivors has been increasing at a rate of 2% per year since 1971 [3]. Alongside this trend, the incidence of second primary tumors has also risen. This condition, known as multiple primary cancers (MPCs), affects about one in six cancer patients [4].

MPCs are defined based on criteria proposed by Warren and Gate, requiring a confirmed pathological malignancy, distinct histological features, and exclusion of metastasis, recurrence, or spread from other tumors [5,6]. In summary, MPCs refer to the presence of more than one tumor with distinct origins in a single patient [7]. These cancers can be categorized as synchronous or metachronous depending on whether the interval between tumor reports exceeds six months. Due to variations in population, definition criteria, and follow-up durations [8,9], the incidence of MPCs varies from 2% to 17% [10,11]. Identifying MPCs is crucial for tumor staging and subsequent treatment choices. However, the current clinicopathological criteria for clinical application are limited and insufficient [12], differentiating them from recurrences or metastases is challenging. Epidemiological studies have established that the development of MPCs is a multifaceted process, affected not just by the late effects of treatments like radiotherapy and chemotherapy but also by a spectrum of other factors [4]. These factors encompass lifestyle choices such as smoking and excessive alcohol consumption, environmental exposures, host factors like obesity [13,14], and the individual’s immune and hormonal statuses [15]. Crucially, genetic predispositions and their interactions also play a significant role. Moreover, the risks for specific MPCs can be influenced by the age at exposure and the age reached (Figure 1). In past decades, research about susceptibility genes faced obstacles due to limitations in bioinformatics technology. Risk genes were categorized as high or medium-risk based on their penetrance and prevalence, taking advantage of family aggregation data. With the advancements in next-generation sequencing (NGS) technology, defining MPCs from a genetic perspective has emerged as a central focus. Cost-effective NGS panels, combined with broader testing criteria, have produced vast amounts of genetic data. Statistical genetics also made finding correlations between tumors achievable. Researchers found that three main genetic factors contributing to MPCs development: the mutations targeting oncogenes and caretaker genes directly involved in MPCs, the potential influence of pleiotropic loci spanning various cancer types, and the regulatory impact of deleterious mutations following treatment.

In this review, we will discuss these susceptible variants and try to explain their functions according to the signaling pathway background. The association network between genetic signatures and different tumor pairs will also be summarized.

## 2. The Mutations Targeting Oncogenes and Caretaker Genes in MPCs

The discovery of mutations in DNA repair genes in hereditary cancers provides significant evidence in favor of the mutator hypothesis, which suggests that germline mutations in classic caretaker genes are responsible for genomic instability. Moreover, oncogene-induced DNA replication stress contributes to genomic instability and the subsequent development of cancers [16].

### 2.1. MSI/MMR in MPCs

Microsatellites are short, simple, repeating DNA sequences scattered throughout the genome, with some sequences repeating up to 100 times. As a result of the propensity for strand slippage during DNA replication, they are intrinsically hypermutable. To maintain regular cell activity, the production of insertion/deletion loops requires the DNA mismatch repair (MMR) mechanism. If these errors are not adequately repaired, microsatellites may exhibit variable expansion or contraction, which is referred to as microsatellite instability (MSI) [17] (Figure 2A). As a type of genomic instability, MSI serves as a marker of heightened tumorigenesis risk [18,19,20,21]. The abnormalities of MMR have been shown to be strongly related to MPCs, especially in digestive tumors.

#### 2.1.1. Multiple Colorectal Cancers

Multiple colorectal cancers account for five percent of cancer patients [22,23,24,25]. The hypermutation caused by deficient mismatch repair (dMMR) increases the chance for more than one colorectal tumor to arise. Lynch syndrome (LS) is a hereditary disorder closely related to multiple colorectal cancers. This syndrome is characterized by an elevated risk of digestive tumors, especially on the right side of the colon. Moreover, LS often co-exists with other cancers like endometrial, gastric, and other extracolonic cancers [26,27,28,29,30]. Mutations in MLH1 (MutL homolog 1) or MSH2 (MutS homolog 2) account for 60% to 80% of cancers associated with Lynch syndrome [31,32,33,34,35,36,37,38]. The interval of tumorigenesis, survival expectations, and familial heritability are associated with microsatellite stability and MMR function.

The predominant mechanism for MSI in multiple colorectal cancers is either the loss of MLH1 expression or hypermethylation of the MLH1 promoter. Notably, metachronous colorectal cancers tend to exhibit higher rates of MSI compared to synchronous tumors. For MSI-high synchronous tumors, a predominant occurrence of MLH1 promoter hypermethylation is observed. Velayos et al. noted that synchronous colorectal tumors with extensive MLH1 promoter methylation seldom have a family history of cancer, implying a dominant sporadic origin for these tumors. Conversely, MSI in metachronous colorectal cancers is more often due to MLH1 inactivation rather than promoter methylation or MSH2 inactivation, and these mainly have a hereditary pattern [25].

#### 2.1.2. Multiple Gastric Cancers

Multiple gastric cancers (MGC) are characterized by the presence of more than one tumor in the stomach, with incidence rates ranging from 6% to 14% [39,40,41]. The prognosis for patients with MGC is poorer compared with those with solitary gastric cancer [42,43]. Additionally, the SEER database supports the notion that repeated tumorigenesis in the stomach may increase the susceptibility of other organs to malignancy [44]. Thus, physicians should remain vigilant for the existence of additional gastric tumors, particularly in elderly patients who are considered high-risk [45,46].

In cases of MGC, impairment in the MMR system has been observed, leading to a higher incidence of MSI compared with solitary gastric cancers. Conversely, patients with MSI-high also have an elevated risk of developing secondary gastric cancer compared to those with microsatellite stable tumors (MSS) [47]. Wang A performed whole-exome sequencing (WES) on 33 tumor samples from 16 individuals with MGC. They identified MSH2 mutations in all genetic MGC patients, with the germline MSH2 X314_splice being the most frequently observed mutation. Additionally, mutant NCOR2 (Nuclear co-repressor 2), AHNAK (Neuroblast differentiation-associated protein, also known as desmoyokin) [48], and TP53 genes were also noticed in this research, but further research is required to verify their significance [49]. The presence of multiple tumors in gastric cancer patients can also be observed in cases of hereditary diffuse gastric cancer syndrome (HDGC). HDGC is characterized by autosomal dominant mutations in the CDH1 tumor suppressor gene (encoding the protein E-cadherin), leading to increased risks of diffuse gastric cancer and lobular breast cancer [50,51]. For those carrying pathogenic CDH1 variants, a prophylactic total gastrectomy is recommended to manage gastric cancer risk [52].

Considering the central role of the MMR/MSI system in multiple digestive cancers, immunotherapy may be an effective choice for optimizing their treatment [53].

### 2.2. DNA Polymerase Mutations in MPCs

DNA polymerases ε (Pol ε) and δ (Pol δ) play essential roles in DNA replication and repair. The POLE gene, found on chromosome 12q24.3, codes for Pol ε’s largest subunit, while POLD1 encodes the catalytic subunit of Pol δ [54]. The exonuclease domains of POLE and POLD1, which exhibit notable homology, are crucial for maintaining DNA replication accuracy [55] (Figure 2A). Mutations in these genes can significantly impact genomic stability. Both DNA mismatch repair and polymerase proofreading, which can function independently or simultaneously, are vital for genomic fidelity. Recent studies have demonstrated that tumors with genetic alterations in the proofreading domain of DNA polymerases POLE or POLD1 exhibit a high number of mutations [56]. Tumors with dMMR/MSI-H often exhibit high tumor mutation burdens (TMB). POLE/POLD1 mutations, linked to ultrahigh mutation rates, can exacerbate this instability, leading to more severe replication repair defects and increased mutation rates [57,58].

Mutations in POLE, both somatic and germline, have been identified in various types of tumors, including endometrial, ovarian, colorectal, melanoma, bladder, esophageal, and lung cancers [59,60]. Wang et al. [59] reported in a study involving 47,721 cancer patients that POLE and POLD1 somatic mutations were presented in 2.79% and 1.37% of cases, respectively. Notably, POLE mutations are frequently observed in synchronous endometrial and ovarian carcinomas. Ishikawa et al. [61] reported a high incidence of these mutations in such cases. Specifically, the POLE p.S459F hotspot mutation was found in samples of endometrioid endometrial carcinoma from patients with synchronous endometrioid endometrial and ovarian carcinomas, which generally have a better prognosis compared with individual occurrences of these cancers independently [62]. Additionally, F Cao reported a case of POLE mutation in endometrioid carcinoma with concurrent HPV-associated cervical adenocarcinoma [63].

In the development of multiple primary cancers, hypermutation and DNA instability play crucial roles. POLE mutations, in addition to mismatch repair defects causing microsatellite instability, contribute to increased mutation loads and a higher risk of MPCs. POLE/POLD1 mutations are associated with high mutation loads, increased neoantigens, and enhanced immune cell infiltration in tumors. These findings suggested that these mutations may serve as biomarkers to predict the efficacy of immunotherapy. However, additional research is necessary to validate this hypothesis. Interestingly, POLE/POLD1 mutations are also correlated with younger age and better outcomes in patients, indicating their potential as new surveillance indicators.

### 2.3. The RTK-RAS Pathway in MPCs

The RTK-RAS pathway is a fundamental cellular mechanism that governs various physiological processes [64,65,66]. Receptor tyrosine kinases (RTKs) are the cell surface receptors that are specifically activated after binding to ligands, such as epidermal growth factor (EGF). This activation serves as a catalyst for downstream protein activity, which in turn influences cell survival, proliferation, and differentiation (Figure 2B). Specific driver mutations associated with this pathway have been identified in MPCs.

#### 2.3.1. Multiple Primary Lung Cancers

Multiple primary lung cancers (MPLC) are characterized by two or more lung tumors of independent origin emerging either synchronously or metachronously [67,68]. With the advent of high-resolution computed tomography, about 15% of lung cancer patients have been found to exhibit MPLC [69,70]. These are notably common in nonsmoking women under 60, predominantly presenting as ground-glass opacities, adenocarcinomas, and in early stages. MPLC generally have a poorer prognosis compared with solitary lung cancer [70] and exhibit decreased sensitivity to both targeted therapy and immunotherapy [71]. A persistent challenge with MPLC, as with all multifocal tumors, is determining the evolutionary origin of the lesions. Nowadays, researchers are concentrating on the genetic aspects to deepen their understanding.

EGFR (epidermal growth factor receptor), a prominent member of the HER/ErbB family and one of the earliest studied receptor tyrosine kinases (RTK), is the most commonly mutated gene in MPLC [71]. Although EGFR-TKIs (tyrosine kinase inhibitors) have dramatically improved the clinical outcomes of non-small cell lung cancer (NSCLC) patients harboring EGFR-sensitive mutations (exon 19 deletion or exon 21 p.L858R), the mechanism of the poor efficacy of targeted therapy remains unclear. Recent studies found that most MPLC patients had different mutation spectra, but half of them shared ≥1 putative driver gene among their multiple lesions. This finding in multifocal lung cancers was supported by the “convergent evolution hypothesis” that heterogeneous driver mutations in multiple tumors from the same patient aggregate on the same signaling pathway to activate key carcinogenic pathways [72]. Germline mutations in EGFR are rare but may contribute to heritable predisposition and oncogenesis. EGFR p.T790M germline mutation has been previously estimated to be involved in 1~4% EGFR mutant NSCLC patients and hereditary lung cancer. The exon 20 p.R776H germline mutation is of special concern in MPLC, but not detected in any of the family probands. Another EGFR mutation, p.V769M, was also found to be involved in germline-related MPLC with somatic co-mutations in EGFR mutations (p.L861Q or p.L858R) [73]. It has been reported that six loci, comprising seven SNPs, namely rs2736100 at 5p15.33, rs2853677 at 5p15.33, rs2179920 at 6p21.32, rs3817963 at 6p21.3, rs7636839 at 3q28, rs7216064 at 17q24.3, and rs2495239 at 6p21.1, have been linked to the susceptibility of lung adenocarcinomas with EGFR mutation [74].

Over a third of EGFR p.L858R mutations are associated with a high TMB [71], potentially a crucial element in the pathogenesis of MPLC. As the most frequent co-mutation with EGFR p.L858R, TP53 mutations facilitate resistance evolution in EGFR-mutant lung adenocarcinoma. These resistance patterns highlight the adaptability of cellular processes in tumorigenesis, often leading to tumor recurrence and progression to more aggressive forms [75]. Additionally, these mutations could be linked to the development of additional lung cancers. Histologic transformation from NSCLC to squamous cell carcinoma (SCC) or small cell lung cancer (SCLC) has been identified in 2–15% of patients subjected to progression on osimertinib treatment. Complete inactivation of both RB1 and TP53 was strikingly associated with an increased risk of transformation from NSCLC to SCLC among EGFR-mutant lung adenocarcinomas, suggesting the predictive value of these two mutations in SCLC transformation [76]. In addition, epithelial-mesenchymal transition (EMT), another type of histologic transformation, has also been described as a mechanism of osimertinib resistance, which is related to NF-κB and TGFβ2 [77]. The ERBB3 (Erb-B2 Receptor Tyrosine Kinase 3), MAP2K2 (Mitogen-Activated Protein Kinase 2), and ARAF genes in the RTK-RAS pathway have also been identified in the onset of additional lung malignancies. Furthermore, research proves that alterations in the NOTCH and Wnt pathways occur in MPLC instances [78].

Mutations in EGFR are also associated with changes in the immune microenvironment [66]. The reduction in HLA (Human Leukocyte Antigen) diversity and PD-L1 expression might affect immunological infiltration and lead to a diminished response to immunotherapy. Nowadays, increasing studies have been working on decoding the intricate interaction between the tumor cells and the tumor microenvironment in MPLC.

#### 2.3.2. Cowden Syndrome

The RTK/PI3K/AKT is an offshoot of the RTK-RAS pathway. Within this pathway, PTEN (Phosphatase and Tensin Homolog) acts as a tumor suppressor gene, ensuring the orderly regulation of cell survival and proliferation processes [79]. By dephosphorylating PIP3 (Phosphatidylinositol 3,4,5-trisphosphate) to PIP2 (Phosphatidylinositol 4,5-bisphosphate), PTEN directly inhibits the downstream activation driven by PI3K (Phosphoinositide 3-Kinase), thereby limiting the subsequent triggering of cell growth (Figure 2B). Various sporadic tumors have displayed mutations or a loss of PTEN functionality. The observed association between synchronous endometrial and ovarian carcinoma may also be linked to PTEN mutations and loss of heterozygosity (LOH) [80]. Furthermore, its germline mutations can directly lead to high-risk cancer conditions such as Cowden Syndrome [81,82].

Cowden Syndrome (CS) is an uncommon autosomal dominant disease caused by heterozygous germline mutations of PTEN [83]. Individuals diagnosed with CS face an elevated risk of developing tumors in various organs, notably the thyroid, breast, endometrium, colorectal, and even malignant melanomas [84]. Therefore, patients with familial histories should place significant emphasis on tumor screening, particularly for the breast and uterus.

### 2.4. DSB with MPCs

DNA double-strand break (DSB) is a severe type of DNA damage caused by ionizing radiation, solar ultraviolet (UV), or reactive oxygen species (ROS). There are three primary mechanisms for repairing DSB: homologous recombination (HR), non-homologous end joining (NHEJ), and single-strand annealing (SSA). All of them are vital for maintaining genomic integrity and normal function. Several driver genes, including ATM (ataxia telangiectasia mutated protein), CHEK2 (Checkpoint Kinase 2), and BRCA genes, are involved in the HR process. What is more, abnormalities in the DSB repair process significantly increase the risk of multiple cancer development [85] (Figure 2C).

#### 2.4.1. Hereditary Breast and Ovarian Cancer Syndrome

Breast cancer is the most common cancer in women [86]. In recent decades, advancements in early diagnosis techniques have increased the likelihood of surgical intervention and decreased mortality rates. As a result, the longer overall survival of breast cancer patients has dramatically raised their chance of developing a second tumor [87]. In addition, early onset of breast cancer is considered a risk factor for second cancer and poor prognosis [88,89,90]. 

Ovarian cancer is frequently observed in breast cancer survivors, particularly in cases of hereditary breast and ovarian cancer syndrome (HBOC). This syndrome is characterized by the prevalence of breast with ovarian cancer. Individuals with HBOC syndrome also experience a higher incidence of other malignancies, such as melanoma, pancreas, and prostate cancer. The BRCA gene, a pivotal molecular in the HR process, has been proven a susceptibility gene in this context.

The BRCA gene accepts DSB signals via ATM, which then initiates the HR procedure to repair damaged DNA [91]. Homologous recombination, critical for DNA double-strand break repair, is conserved from bacteria to humans [92,93]. In mammalian cells, the BRCA1/2 protein is pivotal in HR, notably in forming RAD51 presynaptic filaments and safeguarding stalled replication forks [94]. In cases of BRCA1/2 gene mutations, RAD52 takes over the repair role through the RAD51-independent single-strand annealing (SSA) pathway, active especially in direct repeat regions [95,96,97] (Figure 2C). Blocking RAD52 activity in cancer cells lacking BRCA1/2 proves fatal [94], underscoring the critical importance of RAD52 in the DNA repair process. However, RAD52-mediated repair is less precise compared with the BRCA1/2-driven homologous recombination process [98]. This reliance on the error-prone RAD52 in BRCA1/2-deficient cells can result in genomic rearrangements and instability, significantly contributing to an elevated mutation burden and advancing cancer progression. Relative to the general population, carriers of mutant BRCA1/2 have a significantly high risk of both breast and ovarian cancer [99,100,101,102,103,104,105,106]. Shih et al. [107] found that BRCA1/2 mutations were present in 42.9% of breast cancer patients with a second primary tumor. This percentage was 22.7% when the second cancer was not ovarian but up to 84.4% if the second primary cancer originated in the ovary. Not only that, other genes such as CDH1 (Cadherin 1), PTEN, STK11 (Serine/Threonine kinase 11), and TP53 were also found to participate in HBOC cases [108]. The intricacies of molecular mutations and their interactions necessitate more extensive research for a deeper understanding.

HBOC is an inherent disorder, and genetic sequencing is recommended for all women diagnosed with breast or ovarian cancer, regardless of their age of onset [109]. International guidelines also propose offering risk-reducing breast excision surgery to all BRCA1/2 mutation carriers at a young age to lessen the further threat.

#### 2.4.2. The CHEK2 Syndrome

CHEK2 (Checkpoint Kinase 2) is another crucial protein in the DNA damage response network in an ATM-dependent manner [110,111,112]. CHEK2 gene is located upstream of the BRAC and TP53 genes, coding the human analog of the yeast Checkpoint Kinases Cds1 and Rad53. It plays a role in cell cycle checkpoint control, halting cell division to allow time for repair. Phosphorylated CHEK2 can trigger apoptosis via either TP53-dependent or TP53-independent ways (Figure 2C). Known as one of the highest prevalence of germline mutations, CHEK2 is now routinely included in diagnostic NGS panels. Analyses have revealed that CHEK2 is a multisite cancer gene with different penetrance, including prostate, kidney, breast, and thyroid cancers [113,114,115,116,117,118,119,120,121]. 

There are four main founder alleles of CHEK2. Three of them are protein-truncating variants (1100delC, IVS2+1G>A, and del5395), and the fourth is a missense variant (I157T) [121]. The susceptibility to tumors also differs in terms of the allele variants. Cybulski et al. [115] analyzed 4008 cases across 13 tumor types and discovered an elevated risk of thyroid, prostate, and breast cancer in carriers of truncating CHEK2 variants [120,122,123,124,125]. In contrast, missense alleles seem more likely to be associated with kidney cancer [109,119,126]. The CHEK2 I157T variation was more prevalent in MPC patients than in those with solitary cancers [118]. These findings indicate a significant increase in CHEK2 mutations among MPC populations. However, using CHEK2 as a predictive marker remains challenging due to its limited penetrance in cancer patients and its high prevalence among control subjects.

### 2.5. p53 Inactivation with MPCs

p53 protein is one of the most crucial targets in tumor suppression, with mutations present in over 50% of cancers [127,128,129]. It serves as an intersection for numerous signaling pathways and is instrumental in cell checkpoint and cycle regulation [130,131]. More than 75% of TP53 alterations result in its inactivation, prompting damaged cell survival and proliferation and eventually encouraging cancer invasion and metastasis [132]. The mutant TP53 is also related to immune escape and drug resistance (Figure 2D). Despite significant heterogeneity, p53 inactivation-induced malignancy has been proven to follow a consistent and predictable pattern of genome evolution [133]. The subsequent deficient repair and genetic instability are believed to significantly contribute to MPCs.

#### 2.5.1. Li-Fraumeni Syndrome 

Li-Fraumeni syndrome (LFS) is a rare genetic disorder that increases the risk of several cancers in children and young, including sarcoma, brain tumors, and leukemias [134,135,136,137,138]. Approximately half of LFS patients harbor mutant TP53 and inherit it in an autosomal dominant pattern. These carriers have a near 100% penetrance lifetime cancer risk [139]. 

The likelihood of second tumors in LFS is significantly increased if the first tumor appears at an early age [136,140]. A study by Gonzalez et al. found that more than 50% of patients with TP53 mutations experienced multiple primary cancers, compared with only 32% of patients with wild-type TP53. Germline TP53 mutations were also associated with early-onset cancer [136]. Comprehensive analysis of TP53’s role in LFS requires testing to determine the specific mutation type. Some inherited TP53 mutations in LFS are identical to those acquired mutations in malignancies, such as G245S [141,142], while others are exclusive to LFS as germline-specific mutations, such as R337H [143,144]. There is a greater diversity of novel TP53 germline mutations in LFS compared to sporadic cases. And similar reports could serve as potential biomarkers for identifying high-risk families.

#### 2.5.2. Multifocal Esophageal Cancers 

Esophageal cancer, distinguished by its high incidence and mortality rates, often manifests as multiple tumor lesions [145,146]. Based on clonal origins, these lesions can be classified as either multicentric origin (MO) or intramural metastases (IMM). Identifying the origin will influence later surgical scope and treatment decisions, and patients with multicentric origin generally have a more favorable prognosis.

Most multifocal esophageal cancers are commonly located in the middle or lower esophagus. Their presence can be explained by the “field cancerization” hypothesis, which means the nearby epithelium, like the esophagus, lung, and the head and neck, is exposed to a shared carcinogen, especially for cigarettes and alcohol [147,148,149,150,151,152]. Elder men with a long history of smoking and drinking have been classified as a high-risk group. Regarding the notable sex differences in esophageal cancer incidences, research points to complex interactions between various hormones and associated cancer risks [153,154]. Notably, increased levels of Follicle-Stimulating Hormone (FSH) have been associated with a heightened risk of esophageal cancer [155]. In contrast, elevated levels of Luteinizing Hormone (LH) and testosterone appear to correlate with a decreased risk, although this relationship is influenced by factors related to obesity [156]. Additionally, Dehydroepiandrosterone (DHEA) and estradiol seem to play a protective role, potentially lowering the risk of esophageal adenocarcinoma [157]. Similar to breast and prostate cancer, the interaction between steroid hormone pathways and EGFR/HER2 signaling pathways may offer the potential for targeted therapeutic strategies that address both hormonal influences and growth factor receptor activity in esophageal cancer. However, details regarding the role of endocrine factors in multifocal esophageal cancers remain limited. Researchers found that TP53 alterations are risk factors for the pathogenesis of multifocal esophageal cancers, especially for nonsense mutations [158,159]. The accumulation of inactive p53 protein can solidify precancerous lesions and hasten cancer development. Furthermore, larger esophageal tumor lesions often show increased immune cell infiltration, elevated PD-L1 expression, and more active immune and proliferative networks. This heterogeneous immune microenvironment may be influenced by TP53 diversity, leading to varied responses to immunotherapy. Furthermore, similar immune landscapes have been reported in multifocal hepatocellular carcinoma as well [78,160,161].

p53 is almost universal inactivation in malignancy [162]. However, in individuals with wild-type p53 (wtp53), various regulatory mechanisms participate in diminishing the protein’s activity. Among these, MDM2 (mouse double minute 2 homolog) and MDM4/X (mouse double minute 4 homolog) can play as key negative regulators. MDM2 [163], functioning as an E3 ubiquitin ligase, promotes the degradation of p53 via the proteasome. p53 transcriptionally activates MDM2, creating a feedback loop that allows p53 to regulate its own levels. This regulation, characterized by ubiquitination, is crucial for maintaining p53’s stability and function, particularly in response to cellular stress and DNA damage. On the other hand, MDM4/X mainly influences p53’s transcriptional activity [164]. Normally, MDM2 and MDM4/X collaborate to maintain low p53 levels. However, cellular stress, such as DNA damage, disrupts their interaction with p53, leading to its stabilization and the activation of processes like DNA repair, cell cycle arrest, or apoptosis. Yet, in various cancers, the overexpression of MDM2 and MDM4/X, even in cells with wtp53, can counteract this mechanism. Current research is thus focused on restoring the normal anti-cancer function of p53 in both wtp53 and mutp53 cells. This includes developing molecules to disrupt the wtp53–MDM2/MDM4 interaction in wtp53-expressing cancers and reinstating wtp53-like functions in mutp53-expressing cancers [165]. Unlike genetic mutations, the active regulation of protein might provide new insights for assessing MPC risk, particularly in patients with wtp53.

## 3. Pleiotropic Locus with MPCs

In addition to the mutations targeting oncogenes and caretaker genes directly involved in MPCs discussed above, increasing genetic links between various cancers have been identified due to the explosion of genetic data. This shared genetic foundation across diverse tumors is referred to as pleiotropic loci, which include HOXB13 G84E [166], TERT (telomerase reverse transcriptase) gene mutations [167], 8q24 variants [168], and so on. Additionally, specific single nucleotide polymorphisms (SNPs), including rs555607708, rs146381257, and rs1805008, have emerged as germline variants in different cancer groups. These genetic markers present an elevated risk for MPCs, impacting both cancer survivors and individuals with no history of cancer. 

### 3.1. The Role of the ZNF106 Gene

As an RNA-binding protein (RBP), ZNF106 plays a crucial role in post-transcriptional control and insulin receptor signaling. Yue Wu et al. [169] observed that the rare variant rs146381257 in ZNF106 was prominently overrepresented among patients with breast, lung, prostate, and urinary bladder cancers, as well as those with lymphoid neoplasms. This variant was considered a pleiotropic locus that predisposes carriers to a heightened risk of MPCs. The importance of posttranscriptional control in cell regulation explains why diverse tumors can manifest following ZNF106 mutations.

### 3.2. The Role of MYC Gene

The 8q24 region of the human genome is a well-known cancer pleiotropic locus and stands as one of the most frequently referenced genomic regions. Men diagnosed with prostate cancer exhibit an elevated risk of developing thyroid and colorectal cancers, among others [170,171]. Notably, the replicable discovery of variant rs6998061 in the 8q24 locus, associated with increased risk for both prostate and colorectal cancers, underscores the region’s significance [172]. Among the genes in the 8q24 region, MYC is of particular significance. It plays an essential role in cell growth and is the most extensively studied oncogene [173]. Many cancers display dysregulated expression of the MYC gene. Therefore, the functionally defined MYC carcinogen is a leading candidate for deeper investigation of the pleiotropic locus associated with MPCs. What is more, the unique metabolic pathway dependencies of MYC-driven cancers offer promising therapeutic strategies for MPCs [174]. 

### 3.3. The Role of the TERT-CLPTM1L Gene

Telomeres act as safeguards at the ends of chromosomes, ensuring genomic stability and chromosomal integrity. Telomerase synthesizes telomeric DNA, with TERT serving as one of its primary catalytic components. Under normal conditions, the activity of telomerase is kept in check due to the tight regulation of the TERT gene. However, during carcinogenesis, the induction of TERT and the activation of telomerase prevent telomere shortening. This confers cancer cells with an immortal and aggressive nature [175]. The TERT gene has a neighboring region named CLPTM1L on 5p15.33 [176]. Genome-wide association studies (GWAS) indicate that TERT-CLPTM1L region variants increase the risk of various cancers [167,177]. And the rs401681 variant is associated with heightened susceptibility to lung, bladder, esophagus, prostate, and pancreas cancers [178,179,180]. Furthermore, mutations in TERT often lead to poorer prognosis. Telomere attrition is a hallmark of cellular senescence [181]. The pleiotropic loci in telomerase are of significant concern in cross-cancers.

### 3.4. Another Pleiotropic Locus with MPCs

Tumorigenesis is a multistep process with various pleiotropic loci playing functions in multiple cancers. The SAMHD1 locus, for instance, displays its effect by facilitating proper DNA synthesis and repair [182]. Similar to the BRCA gene, individuals with NCBP1 mutations have an increased likelihood of developing a second tumor in breast cancer survivors. However, BRCA1/2 carriers predominantly develop ovarian cancer, while NCBP1 carriers are more predisposed to cervical cancer [172]. Tumor stem cells are always at the center of MPC research due to their inherent tumorigenicity. ALDH1, a most characteristic tumor stem cell marker, is markedly expressed in malignancies like breast, lung, colorectal, liver, gastric, and head and neck tumors [183]. TET2 locus variations, which play a role in DNA methylation, have also been observed to enhance the chance of prostate, ovarian, breast, and colorectal cancers [184,185,186,187]. Additionally, genetic mutations in inflammation and its sub-pathway are essential for the development of lung, colorectal, ovarian, and breast cancer [188].

Recent research into pleiotropic loci has significantly enhanced our understanding of the genetic signatures of MPCs. Theoretically, without the constraints of human lifespan, an individual could potentially experience multiple tumors influenced by pleiotropic loci over time. However, it is crucial to note that not all observed intragenic mutations, particularly single nucleotide variations, have a proven impact on protein structure and function. Furthermore, a substantial number of these mutations may not be involved in tumorigenesis; they could be passenger mutations rather than drivers [189]. Beyond coding and non-coding mutations, other driver alterations, such as structural variants and epigenetic modifications, also play a role in tumorigenesis by offering selective advantages to certain cells [190]. Advancements in sequencing technology have shed light on more and more gene expression stages in the regulation of MPCs, including the overexpression of oncogenes, underexpression of tumor suppressor genes, post-transcriptional control, and DNA methylation. Exploring other dimensions of tumors through transcriptomics, proteomics, and methylomics, along with systematic assays of gene function and interactions and single-cell profiling, is crucial for expanding our knowledge [189,191,192]. Bridging the gap between the list of genes identified in sequencing reports and a comprehensive understanding of tumorigenesis remains a significant challenge in cancer genomics. The need for more standardized and advanced studies to validate these findings is imperative.

## 4. Risk Modification of Mutations in Treatment-Exposure-Related MPCs

With the survival rates increased, physicians noticed some genetic alterations can regulate the risk of secondary cancer following initial treatment [193]. For instance, Hodgkin lymphoma (HL) patients with detrimental mutations frequently develop subsequent non-Hodgkin lymphoma, acute lymphoblastic leukemia, and ovarian cancers after conventional treatment. These secondary cancers mainly contribute to non-relapse mortality among HL patients [194,195,196,197,198,199,200]. Additionally, interventions such as chemotherapy and radiotherapy can exacerbate genetic abnormalities. Individuals with highly penetrant genetic mutations, like those in RB or TP53 genes [201,202,203,204], are often advised to avoid radiotherapy due to the increased risk of secondary cancers in irradiated areas [140,205]. Epidemiological studies have reported increased risks of second lung cancers after breast cancer irradiation, potentially attributed to smoke or the delivered dose to the lung [206,207]. Tamoxifen used for clinical prevention and treatment of breast cancer has been associated with a significantly increased risk of endometrial cancer [208]. A systematic review of 21 studies evaluating the risk of secondary malignancies in men with prostate cancer who received radiation therapy found an increased risk of bladder cancer and colorectal cancer [209]. However, the genetic mechanisms driving these associations remain largely unexplored. Certain treatments for primary tumors have also been associated with reduced risks of secondary cancers. For instance, aspirin usage in HNPCC patients [210], selective estrogen receptor modulators (SERMs) for breast cancer cases [211,212,213,214], and isotretinoin for head and neck malignancies [215] have all shown such potential, but the exact mechanisms behind these risk reductions remain understudy.

## 5. Future Research Direction 

To comprehensively understand susceptibility genes, researchers employ two primary methods. The first involves candidate gene analyses, which focus on a selection of biologically relevant genes/pathways. The second method is GWAS, utilizing DNA arrays capable of identifying over 1 million SNPs. While candidate gene approaches are hypothesis-driven, genome-wide strategies provide a broader, agnostic exploration of the genetic landscape. A combination of both methods is crucial for hereditary explorations. However, it is important to note that this work can be cost-intensive because of the long-term follow-up and extensive genetic sequencing tests. Therefore, careful consideration should be given to protocol design, including accurate sample size determination, precise phenotype definition, and the availability of high-quality DNA samples to maximize the utility of research efforts.

Research on multiple primary cancers is currently in its early stages. It is important to acknowledge that the prevalence of MPCs may have been underestimated due to limited follow-up time. To advance this field further, collaboration among clinicians, researchers, and patient advocates is crucial. The National Cancer Institute (NCI) has identified key research priorities for MPCs, which encompass the following: (1) establishing national research facilities dedicated to cancer survivor studies; (2) organizing a collaborative network for biospecimen collection; (3) developing informational and technical support; (4) renewing epidemiological methods; and (5) creating evidence-based guidelines for clinical practice [216]. Establishing more efficient and scientific study cohorts based on existing knowledge is the focus of future research.

## 6. Conclusions

Our analysis of inherited traits associated with multiple primary cancers has identified three key factors contributing to their development: the mutations targeting oncogenes and caretaker genes directly involved in MPCs, the potential influence of pleiotropic loci spanning various cancer types, and the regulatory impact of deleterious mutations post-treatment. Tumors develop through the accumulation of mutations over time. Among these, many are “passenger” mutations, which do not actively contribute to the cancerous process. In contrast, “driver” mutations are those that provide a selective growth advantage to tumor cells. Although the impact of positive selection is consistent across all driver genes within a tumor cohort, these driver mutations tend to occur at various stages of tumor evolution [189]. The process of cell transformation is further complicated by other gene expression mechanisms, including epigenetic modifications and post-transcriptional regulation. This continuous positive selection becomes particularly relevant in the sequential development of multiple primary cancers, where driver alterations may emerge under new conditions, such as treatment responses, drug resistance, or genomic instability. Yet, regardless of the diversity of these alterations, they universally influence three fundamental cellular processes: cell fate determination, cell survival, and genome maintenance [217].

Our investigation has revealed distinctive correlations between various genetic alterations and the occurrence of specific MPC combinations (Table 1). Notably, deficiencies in mismatch repair and the presence of MSI tend to affect digestive cancers (Figure 2A). POLE/POLD1 mutations, which are closely linked to the crucial roles of DNA polymerases in DNA replication and repair, can lead to ultrahigh mutation rates. Such mutations in POLE are often observed in cases of synchronous endometrial and ovarian carcinomas (Figure 2A). As one of the most important oncogenes, EGFR alterations in the RTK-RAS pathway consistently manifest in multiple primary lung cancers, while PTEN mutations, as a tumor suppressor gene, within the RTK/PI3K/AKT pathway increase the risk of thyroid, breast, endometrial, colorectal cancer, and malignant melanomas (Figure 2B). Failures in DSB repair, often due to BRCA1/2 and CHEK2 alterations, are also prevalent in MPCs. BRCA mutations, along with the upregulation of SSA activity, may be a compensatory mechanism, and genomic instability caused by the error-prone SSA pathway may contribute to second malignancies arising (Figure 2C). Lastly, p53 inactivation is a frequent phenomenon in multifocal esophageal cancers and others (Figure 2D). Establishing the association network between genetic signals and MPCs pairs can help risk stratification and the surveillance strategies advance. However, more evidence is required to validate these findings.

In conclusion, the current clinicopathological criteria for clinical application are indeed limited and insufficient. Exploring susceptibility genes not only has the potential to improve diagnostic accuracy but also enables more precise risk stratification. Multiple primary cancers are underutilized resources for delving into the origins of tumors. Consequently, there is a pressing need for larger and more rigorous research endeavors to enhance our understanding of the genetic background of MPCs.

## Figures and Tables

**Figure 1 cancers-15-05788-f001:**
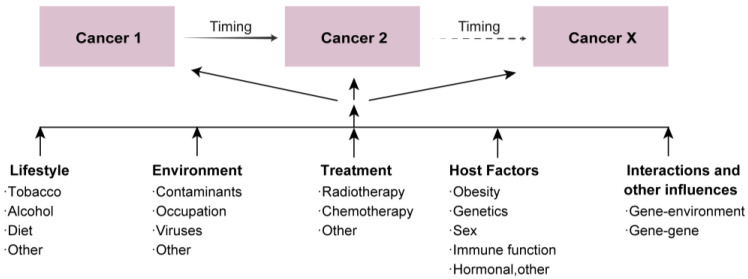
The risk factors of multiple primary cancers development.

**Figure 2 cancers-15-05788-f002:**
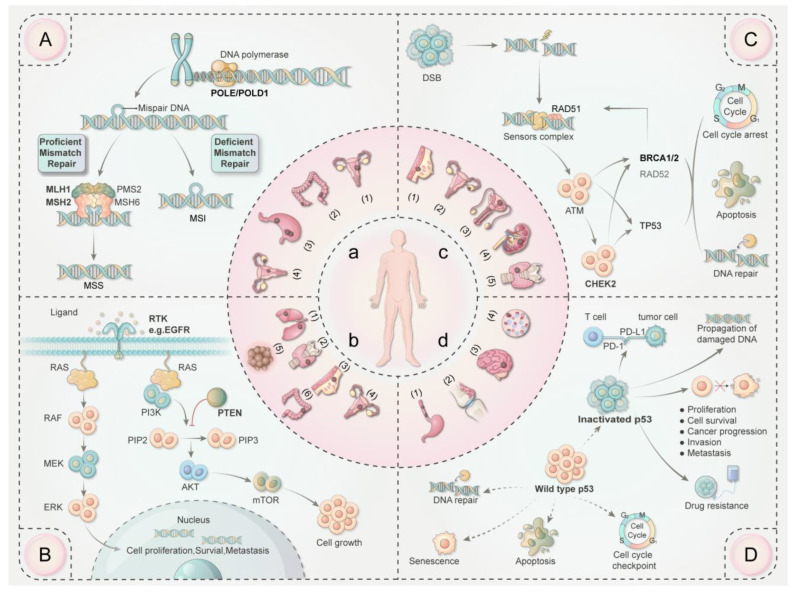
The association network between genetic signatures and different multiple primary cancer pairs. (**A**) The DNA mismatch repair mechanism. (**B**) The RTK/RAS and RTK/PI3K/AKT pathway. (**C**) The homologous recombination mechanism for DNA double-strand break repair. (**D**) Difference between wild-type p53 and mutant p53 on their function and oncogenic activity. a: (1) Endometrial cancer. (2) Multiple colorectal cancers. (3) Multiple gastric cancers. (4) Synchronous endometrial and ovarian cancers. b: (1) Multiple primary lung cancers. (2) Thyroid cancer. (3) Breast cancer. (4) Endometrial cancer. (5) Malignant melanomas. (6) Colorectal cancer. c: (1) Breast cancer. (2) Ovarian cancer. (3) Prostate cancer. (4) Kidney cancer. (5) Thyroid cancer. d: (1) Multifocal esophageal cancers. (2) Sarcomas. (3) Brain tumors. (4) Leukemias. Abbreviation: MSI: microsatellite instability. MSS: microsatellite stability. RTK: receptor tyrosine kinases. RAS: rat sarcoma protein. RAF: rapidly accelerated fibrosarcoma protein. MEK: mitogen-activated protein kinase. ERK: extracellular signal-regulated kinase. PI3K: phosphatidylinositol 3-kinase. PIP2: phosphatidylinositol 4,5-bisphosphate. PIP3: phosphatidylinositol 3,4,5-trisphosphate. PTEN: phosphatase and tensin homolog. AKT: protein kinase B. mTOR: mammalian target of rapamycin. DSB: DNA double-strand break. HR: homologous recombination. ATM: ataxia telangiectasia mutated protein. CHEK2: Checkpoint Kinase 2. BRCA1/BRCA2: breast cancer gene 1/2. PD-1: programmed cell death protein 1. PD-L1: programmed cell death 1 ligand 1.

**Table 1 cancers-15-05788-t001:** Examples of genetic mutations and syndromes or pleiotropic loci that increase the risk of developing multiple primary cancers.

Gene	Syndrome	Classification	Process	Major Component Cancers
MLH1, MSH2	Lynch syndrome	TSG *	Genome maintenance	Multiple colorectal cancers
				Endometrial cancer
				Multiple gastric cancers
CDH1	Hereditary diffuse gastric cancer syndrome	TSG	Cell fate	Diffuse gastric cancer
				Breast cancer
POLE, POLD1	-	-	Genome maintenance	Endometrial cancer
				Ovarian cancer
				Colorectal cancer
				Malignant melanoma
EGFR	-	Oncogene	Cell survival	Multiple primary lung cancers
PTEN	Cowden Syndrome	TSG	Cell survival	Endometrial cancer
				Ovarian cancer
				Thyroid cancer
				Breast cancer
				Colorectal cancer
				Malignant melanoma
BRCA1/2	Hereditary breast and ovarian cancer syndrome	TSG	Genome maintenance	Breast cancer
				Ovarian cancer
CHEK2	-	TSG	Genome maintenance	Thyroid cancer
				Prostate cancer
				Breast cancer
				Kidney cancer
TP53	Li-Fraumeni syndrome	TSG	Cell survival	Sarcoma
				Brain cancer
				Leukemias
				Multifocal esophageal cancers
**Gene/** **Chromosome**	**Pleiotropic locus**	-	-	**Major component cancers**
ZNF106	rs146381257	-	-	Breast cancer
				Lung cancer
				Prostate cancer
				Bladder cancer
Chromosome 8q24	rs6998061	-	-	Prostate cancer
				Colorectal cancer
				Thyroid cancer
TERT	rs401681	-	-	Lung cancer
				Bladder cancer
				Esophagus cancer
				Prostate cancer
				Pancreas cancer

* TSG: Tumor Suppressor Gene.

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
