# Peer review of "Susceptibility Genes Associated with Multiple Primary Cancers"

_cancers, 2023, doi:10.3390/cancers15245788_

Round 1

Reviewer 1 Report

Comments and Suggestions for Authors

The review manuscript is written well. Minor comments:

1. Please add a Table to summarize the genes and mutations associated with each MPC.

2. Since MDM2 and MDM4/X are the major regulator of p53, any MDM2 and MDM4/X info associated with MPCs which are discussed in the current manuscript?

Reviewer 2 Report

Comments and Suggestions for Authors

Authors looked into analysis of multiple primary cancers (MPCs) and attempted to develop a tool which allows to distinguish MPCs from recurrences or metastases. Authors discussed three primary inherited factors contributing to MPCs, including the sustained activation of conventional oncogenes, the pleiotropic loci spanning various cancer types, and the regulatory impact of deleterious mutations following treatment. The susceptibility genes were considered. The association network between genetic signatures and different tumor pairs was also assessed. The review is interesting and well-written.

 There are only a couple of issues to address.

1.      All used abbreviations should be deciphered. For instance EGFR  etc was not deciphered. For instance, NCOR2 ( nuclear co-repressor 2) should be deciphered and proper citation/reference indicated.

2.      Citation are missing for some important statements ( line 112: “…mutant NCOR2, AHNAK, and 111 TP53 genes were also noted in certain instances,”… which instances? Reference? The text should be checked properly and relevant citations should be included.

3.      Role of obesity as a contributing factor to MPCs should be accented and appropriate sections/pararaphs should be included for various cancer types. See this reference: Renehan AG, Tyson M, Egger M, et al.. Body-mass index and incidence of cancer: a systematic review and meta-analysis of prospective observational studiesLancet 2008;371:569–78. And/or this ref: Hahn KM, Bondy ML, Selvan M, et al.. Factors associated with advanced disease stage at diagnosis in a population-based study of patients with newly diagnosed breast cancerAm J Epidemiol 2007;166:1035–44.

4.      Prostate cancer may coincide with colon cancers ( metachronous association, see this citation ; Tanjak P, Suktitipat B, Vorasan N, Juengwiwattanakitti P, Thiengtrong B, Songjang C, Therasakvichya S, Laiteerapong S, Chinswangwatanakul V. Risks and cancer associations of metachronous and synchronous multiple primary cancers: a 25-year retrospective study. BMC Cancer. 2021 Sep 23;21(1):1045. doi: 10.1186/s12885-021-08766-9.

And/or this ref: Fan CY, Huang WY, Lin CS, Su YF, Lo CH, Tsao CC, Liu MY, Lin CL, Kao CH. Risk of second primary malignancies among patients with prostate cancer: A population-based cohort study. PLoS One. 2017 Apr 6;12(4):e0175217. doi: 10.1371/journal.pone.0175217. PMID: 28384363; PMCID: PMC5383246.). A strong synchronous association was found between uterine and ovarian cancers. This should be pointed out.

5.       Esophageal cancer: role of steroid hormones should be accented. See this references; Sukocheva OA, Li B, Due SL, Hussey DJ, Watson DI. Androgens and esophageal cancer: What do we know? World J Gastroenterol. 2015 May 28;21(20):6146-56. doi: 10.3748/wjg.v21.i20.6146. And/or this ref:  Yang H, Sukocheva OA, Hussey DJ, Watson DI. Estrogen, male dominance and esophageal adenocarcinoma: is there a link? World J Gastroenterol. 2012 Feb 7;18(5):393-400. doi: 10.3748/wjg.v18.i5.393.) Role of steroid hormones and a link to EGFR/HER2 network should be accented.

6.       Conclusion should be more focused on the identified molecular targets.

7.       A conclusive diagram with the most important molecular targets should be created and included as a final figure ( Figure 3) and/or as a graphical abstract.

Comments on the Quality of English Language

The English is fine, only minor editing/checking is required.

Reviewer 3 Report

Comments and Suggestions for Authors

In this review, the authors summarize genetics of multiple primary cancers. The authors touch on important points but I recommend some improvements below.

In figure 1 there is some misspelling in the legend: “cancers” not “caners”. Also, the figure shows various exogeneous and endogenous risk factors, but age is listed as host factor. I suggest that age be in its own category as it is a main driver in non-hereditary cancers. Additionally, another factor that’s missing from the figure is treatment. Many cancer treatments (especially radiation) can cause secondary cancers. Multiple tumors are often present in patients previously treated with radiation or chemotherapy. In fact, one of the reasons for a push towards single molecule (targeted therapy) is to prevent unwanted side-effects of treatments.

Section 2. The authors discuss mismatch repair problems which is indeed a driver for these cancers. Mutations in mismatch repair genes can cause multiple tumors. However, what is missing is a discussion on polE frameshift mutation signature which increases mutation burden and causing multiple cancers. The authors should talk about polE and epistatic interaction between mutation in this replicative polymerase gene and mismatch repair genes. Additionally, the authors miss endometrial and ovarian cancers which are driven by polE and mismatch repair mutations. Please devote a paragraph on these cancers as well.

Section 3. The authors allude to TKIs drugs which inhibit EGFRs but only briefly. Please expand a discussion on how adaptation to some of these TKIs can cause activating mutations in EGFR that can lead to other cancers. I suggest perhaps referencing this paper: PMID: 32283832. This relates to figure 1 and my point there that cancer treatments can cause secondary tumors.

Section 4. The DSB description should also be expanded a bit. The authors do start with a general theme on DSB repair but focus on BRCA genes and CHEK2. A paragraph should be included on the general role of DSB repair and how mutations in DSB genes can cause multiple cancers. For example, BRCA2 mutants rely on RAD52, and it has been demonstrated that RAD52 increases the mutation burden because the gene is not as good as BRCA2 at repairing DSBs. Thus, patients harboring BRCA mutations will almost always develop multiple cancers. The authors should expand with a discussion on how mutations in DSB repair pathway can cause multiple cancers. See PMID: 36107942 and references therein.

Section 6. The authors classify the genes discussed in previous sections as “driver oncogenes”. There are two problems with this. First not all genes discussed are oncogenes (e.g. BRCA genes are tumor suppressors). Second, not all may be driver. They may cause cellular transformation but not necessarily be “driver”. Please see PMID: 32778778 for discussion on what constitutes “driver” genes. But the authors do make a good point that mutations in driver genes are likely to cause multiple cancers. I suggest including a paragraph on mutations in driver vs passenger genes.

In section 6, the authors also talk about changes in gene expression which is unlike mutation. Whereas mutation almost always inactivates the gene, changes in gene expression does not affect the structure of the protein. Changes in gene expression is a major cancer driver process. The authors may want to dedicate a paragraph on explaining how gene expression changes can cause multiple cancers.

Comments on the Quality of English Language

English is OK

Round 2

Reviewer 3 Report

Comments and Suggestions for Authors

The authors have made significant changes that greatly improve the manuscript. This reviewer is satisfied.